# Hummingbird plumage color diversity exceeds the known gamut of all other birds

Gabriela X. Venable [1,2✉], Kaija Gahm [1,3] & Richard O. Prum [1✉]

A color gamut quantitatively describes the diversity of a taxon's integumentary coloration as seen by a specific organismal visual system. We estimated the plumage color gamut of hummingbirds (Trochilidae), a family known for its diverse barbule structural coloration, using a tetrahedral avian color stimulus space and spectra from a taxonomically diverse sample of 114 species. The spectra sampled occupied 34.2% of the total diversity of colors perceivable by hummingbirds, which suggests constraints on their plumage color production. However, the size of the hummingbird color gamut is equivalent to, or greater than, the previous estimate of the gamut for all birds. Using the violet cone type visual system, our new data for hummingbirds increases the avian color gamut by 56%. Our results demonstrate that barbule structural color is the most evolvable plumage coloration mechanism, achieving unique, highly saturated colors with multi-reflectance peaks.

[1] Department of Ecology and Evolutionary Biology, and Peabody Museum of Natural History, Yale University, New Haven, CT, USA. [2] Present address: Department of Evolutionary Anthropology, Duke University, Durham, NC, USA. [3] Present address: Department of Ecology and Evolution, Univeristy of California, Los Angeles, California, CA, USA. ✉email: gabriela.venable@duke.edu; richard.prum@yale.edu

oloration has many functions in the lives of animals, including sexual and social communication, crypsis, aposematism, thermoregulation, and more[1]. The phenomenon of animal coloration involves a cascade of physical and biological events beginning with the transmission of ambient light through an animal's habitat and incident on its body, interacting with pigments and optical nanostructures in the animal integument, and reflecting back into the environment. The sensory component occurs when this light is transmitted through another individual's eye and is absorbed by ocular pigments, leading finally to a perception of color. The field of sensory ecology encompasses research on all aspects of this cascade of phenomena[1].

An important complement to traditional autecological approaches to sensory ecology, which focus on the sensory challenges and communication mechanisms of individual species, is comparative methods that investigate the evolutionary radiation in color production and sensory systems among species. From this perspective, a color gamut refers to the achieved color diversity of a species, clade, or guild as quantified in the taxonomically appropriate color space. The color gamut of a clade is a consequence of selection acting on integumentary colors and color patterns with superior functions, and the physical, biochemical, and developmental mechanisms that facilitate, bias, and constrain the possible diversity in integumentary colors[2]. Examples include the pelage color gamut of primates as perceived by di- and trichromatic primate species[3] and the plumage color gamut of a comprehensive sample of all living birds using a tetrahedral color space based on avian retinal physiology[2].

Among the most colorful groups of animals, birds have been studied extensively in sensory ecology[1]. In order to properly understand how birds see the world and themselves, theoretical models of bird color vision must consider that birds have tetrachromatic vision; birds possess four retinal color cone types, which allow birds to see ultraviolet/violet (*uv/v*) wavelengths in addition to red (long, *l*), green (medium, *m*), and blue (short, *s*) wavelengths[4]. To quantify avian color perceptions, researchers have employed 3D tetrahedral avian color space models[5,6] that represent all the colors birds can see and distinguish. A color gamut is then defined as the minimum volume covered by a set of color points in an appropriate color space.

Stoddard and Prum[2] used a tetrachromatic color space based on avian retinal physiology to estimate the total avian color gamut for different avian visual systems. Based on a broad sample of 965 reflectance spectra from plumage patches belonging to 111 species of birds selected to encompass the known diversity of avian color production mechanisms, Stoddard and Prum[2] documented that the avian color gamut occupies an unexpectedly small portion of the avian color space– between 26% and 30% of the total differentiable colors (for the avian violet-sensitive, VS, and ultraviolet-sensitive, UVS, visual systems, respectively). Furthermore, they established that pigments comprise a much smaller volume of the avian color gamut than do structural coloration mechanisms, particularly barbule structural colors which are created by constructive interference from layers of melanosomes in the feather barbules[2]. A comparative phylogenetic analysis indicated that novelties in color production mechanism– including the evolutionary origin of new pigmentary and structural coloration mechanisms– have evolved in various avian lineages and expanded the color gamut achieved by those clades[2].

Stoddard and Prum sampled 96 plumage patches with barbule structural coloration including reflectance spectra from three hummingbird species. Given the overwhelming contribution of barbule structural colors (~35%) to the avian color gamut, Stoddard and Prum[2] predicted that a more comprehensive sample of barbule structural colors would likely expand the avian

color gamut substantially, particularly with saturated blue and green hues. Saturated blue and green plumage hues are relatively rare in birds because they require the production of narrow reflectance peaks that stimulate only the small or medium wavelength cones, which have very closely-spaced spectral sensitivies[4,7]. However, barbule structural coloration mechanisms can achieve more saturated colors than avian pigments[8]. By controlling the average thickness of the nanostructures, barbule structural colors can theoretically achieve a complete spectral diversity of saturated hues from ultraviolet to red.

To better estimate the contribution of feather barbule structural colors to the avian plumage gamut, we investigated the plumage color gamut of the hummingbird family (Trochilidae), which is well known for the diversity and complexity of its barbule structural coloration[9–14] (Fig. 1). Using reflectance spectra from over 1600 plumage patches from 114 species of hummingbirds from 68 genera (encompassing one third of all species and 60% of genera in the family), we applied a tetrahedral model of avian color space to: (1) quantify the total plumage color gamut and hue distribution of hummingbirds, (2) revise estimates of the contribution of hummingbirds and barbule structural colors to the total avian plumage color gamut, and (3) revisit the hypothesis that different plumage coloration mechanisms bring different constraints on achievable color diversity. We found that the hummingbird plumage color gamut fills more than a third of avian color space and exceeds that previously documented for all of birds. Additionally, we show that the plumage patches typically used in social communication have larger color gamuts, and we hypothesize that hummingbird plumage coloration has evolved to create more saturated colors within the hummingbird color visual system.

## Results

**Hummingbird plumage color gamut**. The hummingbird plumage coloration gamut occupies 34.2% of the total avian VS cone type color space (Table 1 and Fig. 2; see Fig. 3 for explanation of tetrachromatic avian color space model). The barbule structural colors and melanin patches (i.e., phaeomelanin or eumelanin patches with absence of structural coloration) of hummingbirds occupy 34.0% and 0.775% of the VS color space, respectively (Table 1; Supplementary Fig. 1). Unsurprisingly, hummingbird white (unpigmented) colors contribute the least to the hummingbird color gamut, 0.0148% of VS color space (Table 1; Supplementary Fig. 1). In contrast to previous analyses of avian color diversity[2,14,15], the hummingbird color gamut occupies a larger volume in the VS color space (34.2%) than in the UVS color space (29.6%) (Table 1; Supplementary Fig. 2).

When analyzing resampled subsets of our 114 species of hummingbird, average color gamut volume for subsets of 90 species or greater approaches the value of our full data set (Supplementary Fig. 3 and Supplementary Table 1). Likewise, the standard deviation of gamut volume becomes increasingly negligible as species sample size approaches that of our study (coefficient of variation = 0.0455 at 110 species, *n* = 10 data subsets; Supplementary Table 1). This result indicates that our species sample is more than sufficient in size. However, it also demonstrates the importance of individual species to our final result. All outliers below the average color gamut for each species subset size are those samples that excluded *Boissonneaua jardini*, the hummingbird species with the largest achieved species color gamut volume (see species color gamut volumes below in Table 2).

Different plumage regions in hummingbirds achieve distinct color gamut volumes (Table 3; Supplementary Fig. 4), indicating differences in the history of social, sexual, and natural selection

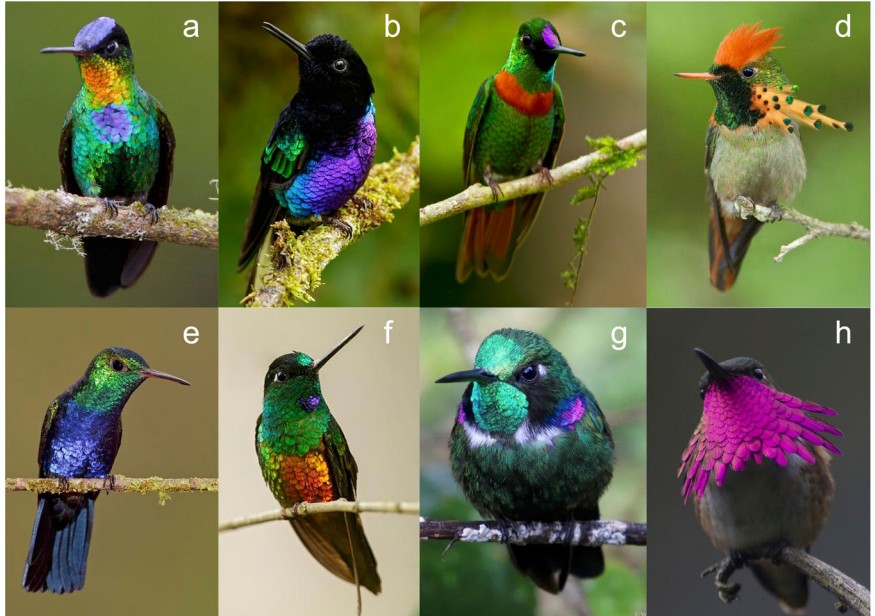

**Fig. 1 Diversity of hummingbird plumage color.** Photographs of males of eight of the 114 hummingbirds analyzed, showing the incredible diversity of hummingbird plumage color, including saturated blues, greens, and purples. **a** *Panterpes insignis*, **b** *Boissonneaua jardini*, **c** *Heliodoxa aurescens*, **d** *Lophornis ornatus*, **e** *Juliamyia julie*, **f** *Coeligena bonapartei*, **g** *Schistes geoffroyi*, and **h** *Atthis ellioti*. Photos reproduced with permission of: **a–f** Glenn Bartley; **g** Wilmer Quiceno; **h** John Cahill.

on these patches (See Discussion). The largest patch gamuts are those for crowns and gorgets (23.1% and 31.0% of VS color space, respectively) while the two smallest patch gamuts are those of undertail coverts (3.65% of VS color space) and wings (2.62% of VS color space; Table 3 and Supplementary Fig. 4).

**Contribution of the hummingbirds to the total avian color gamut.** The breadth of the complete hummingbird color gamut exceeds (VS) or is equivalent to (UVS) the previous estimate of the total color gamut of all living birds[2] (Table 1). Our resampling shows that the VS hummingbird color gamut on average exceeds that of the previous estimate of all birds even when only including 60 species of hummingbird (Table 1, Supplementary Fig. 3; Supplementary Table 1). Hummingbirds expand the known avian color gamut by occupying areas of the avian color space that have not been previously documented. The color gamut and Robinson projections of the hue of hummingbird colors reveal that hummingbird colors densely occupy the *m* (green), the *m–s* (green-blue), and the *s* (blue) regions of the color space and hue map (Fig. 2). Notably, there are several hummingbird spectra that occupy the *s + l* (true purple) wavelength region of VS avian color space, which was sparsely occupied in Stoddard and Prum (Fig. 2)[2].

When the hummingbird data is combined with Stoddard and Prum's data[2], the revised avian color gamut occupies a total of 40.5% of the total VS color space (Table 1 and Fig. 2), which constitutes a 56.0% increase in known gamut volume. From another perspective, the hummingbird color gamut comprises 84.5% of the revised, total known avian color gamut (Table 1). The expansion of the known avian color gamut by hummingbirds is entirely due to the contributions of structural barbule colors. The estimated volume of the avian barbule structural color gamut increased by over 270%, or over three-fold (depending on visual system type), from Stoddard and Prum's[2] estimate– an increase from 9.2% to 34% of the VS color space. In fact, the hummingbird structural barbule color gamut alone exceeds the previously known avian VS plumage gamut of all birds, and is equivalent to

the total known avian UVS plumage gamut reported by Stoddard and Prum[2] (Table 1; Supplementary Table 2).

**Species color gamuts.** The color gamuts of individual hummingbird species vary tremendously from a maximum of 13.8% of the VS color space in the Velvet-purple Coronet, *Boissonneaua jardini* (Fig. 1b) to a minimum of 0.0725% for the Giant Hummingbird, *Patagonia gigas*, which mostly has melanin plumage colors (Supplementary Data 1; Supplementary Fig. 5). The average species color gamut volume is 2.04% of the VS color space. Nine hummingbird species have larger color gamuts than the largest species plumage color gamut previously documented[2]: the Papuan Lorikeet (*Charmosyna papou*) with a color volume of 4.67% in UVS color space (Table 2). Thus, not only are hummingbirds an extremely color-diverse family, but they include the nine most diversely colored species of birds known. However, this comparison may be biased for several reasons. Some methodological differences may contribute to the differences in these results from those of Stoddard and Prum. We made many more reflectance measurements per bird (18+ measurements) than Stoddard and Prum[2] (6–13 measurements). Furthermore, to preserve the natural diversity in coloration due to iridescence, we did not average together the multiple reflectance spectra per patch.

**Notable hummingbird colors.** By examining outliers both in the hummingbird color gamut and in the previously known avian color gamut[2], we identified unusual reflectance spectra that contribute to novel plumage colors among birds (Figs. 4, 5). For example, the gorget of *Boissonneaua jardini* is notable because it is the first barbule structural color reflectance spectrum that we know of with a single, saturated peak in the *uv/v* wavelength component ($\lambda_{max}$ 419 nm; Figs. 1b, 4c, 5). Several *Heliangelus* species are known to have substantial reflectance in the near ultraviolet and violet wavelengths[15], but always with extensive reflectance of longer wavelengths creating an ultraviolet-red color. For example, the *Heliangelus viola* gorget spectrum has a

**Table 1 Summary statistics describing the plumage color gamut of hummingbirds and of all birds (these hummingbird data combined with Stoddard and Prum, 2011).**

| | # of Measurements | Volume | % of Color Space | % of Avian Gamut | Average Color span | Max span | Average Hue disp | Max Hue disp | Average chroma |
|---|---|---|---|---|---|---|---|---|---|
| **Hummingbirds** | | | | | | | | | |
| Total | 5000 | 7.41E-02 | 34.2 | 84.5 | 2.21E-01 | 8.19E-01 | 1.06E+00 | 3.14E+00 | 1.93E-01 |
| Total (UVS) | 5000 | 6.42E-02 | 29.6 | 62.6 | 2.33E-01 | 8.58E-01 | 1.04E+00 | 3.14E+00 | 2.11E-01 |
| Structural Barbule | 3376 | 7.36E-02 | 34.0 | 83.9 | 2.56E-01 | 8.19E-01 | 1.16E+00 | 3.14E+00 | 2.35E-01 |
| Melanins | 1189 | 1.68E-03 | 0.775 | 1.91 | 1.06E-01 | 4.67E-01 | 5.56E-01 | 3.07E+00 | 1.18E-01 |
| Eumelanins | 824 | 9.22E-04 | 0.426 | 1.05 | 6.76E-02 | 3.13E-01 | 6.65E-01 | 3.07E+00 | 7.66E-02 |
| Phaomelanins | 365 | 8.30E-04 | 0.383 | 0.946 | 9.36E-02 | 4.32E-01 | 2.49E-01 | 1.35E+00 | 2.12E-01 |
| Whites | 435 | 3.20E-05 | 0.0148 | 0.0365 | 3.51E-02 | 1.66E-01 | 1.82E-01 | 1.11E+00 | 7.26E-02 |
| **Total Aves** | | | | | | | | | |
| S&P | 966 | 5.62E-02 | 26.0 | 64.1 | 2.05E-01 | 9.64E-01 | 1.28E+00 | 3.14E+00 | 1.39E-01 |
| S&P & Hummingbirds | 5966 | 8.77E-02 | 40.5 | 100 | 2.23E-01 | 9.64E-01 | 1.11E+00 | 3.14E+00 | 1.84E-01 |
| S&P Structural Barbule | 98 | 1.99E-02 | 9.19 | 22.7 | 1.84E-01 | 5.67E-01 | 1.27E+00 | 3.10E+00 | 1.39E-01 |
| S&P & Humm. Struc. Barbule | 3474 | 7.38E-02 | 34.1 | 84.2 | 2.55E-01 | 8.19E-01 | 1.17E+00 | 3.14E+00 | 2.32E-01 |

All statistics pertain to VS measures except those for Total UVS. Summary statistics are given for each presumed plumage color mechanism for hummingbirds, and structural barbule colors for all birds. (For equivalent calculations using UVS, see Supplementary Table 2.)

prominent, saturated *uv/v* peak ($\lambda_{max}$ 420 nm) with a second, saturated, bright peak in the red ($\lambda_{max}$ 656 nm; Figs. 4d, 5). Smaller wavelength secondary peaks were found in many species of hummingbird. However, the creation of secondary peaks with such saturated *uv/v* peaks was only found in the spectra of a few other species, including the gorgets of *Chalcostigma stanleyi*, *Oreonympha nobilis*, *Schistes geoffroyi*, and *Metallura phoebe*. Thus, while this type of reflectance spectrum is not restricted to *Heliangelus*, it should still be considered rare. The gorgets of both *Boissonneaua jardini* and *Heliangelus viola* expand the total avian gamut in terms of saturated UV/V colors (Fig. 5; Supplementary Fig. 2). Yet overall, the saturated UV/V portions of avian color space are sparsely occupied by hummingbirds (Fig. 2; Supplementary Fig. 2).

The production of structural colors with multiple harmonic peaks in the visible spectrum allows some hummingbird species to create colors in the green + UV/V region (Figs. 2, 4, 5), formerly recognized as a vacant hue "ocean"[2], and to fully occupy the true purple ($s + l$) region of the color space (Fig. 2). Such distinctive colors include the cheek, crown, and gorget of *Schistes geoffroyi*, which create unique reflectance spectra with three apparently harmonic peaks (cheek and crown shown in Figs. 1g, 4a, g, 5). No other reflectance spectra measured in this study show similar peaks.

## Discussion

The avian plumage color gamut is much more diverse than previously estimated[2]. We demonstrate that hummingbird barbule structural colors contribute substantially to the total color diversity of living birds, occurring in areas of the avian color space that were sparsely occupied in Stoddard and Prum[2], which most notably included saturated blues, greens, and true purples (blue + red). Such regions of the avian color space were suggested to be unoccupied because these colors are challenging to create, rather than because they might function poorly for communication[2]. Our results support this hypothesis because hummingbird coloration densely occupies these regions of the avian color gamut (Fig. 2d), using plumage patches that generally play particularly important roles in hummingbird communication, such as throat and crown plumage patches (Supplementary Fig. 5)[16,17]. The greater color diversity uncovered by our study suggests that barbule structural coloration is the most versatile class of all plumage coloration mechanisms and poses the least constraints on the evolvability of plumage color diversity. Barbule structural colors evolve through changes in the size, shape, spacing, and refractive index of barbule melanosome nanostructures, but little is known about how changes in these parameters themselves evolve[18].

The UV/V + green region of avian color space remains mostly unoccupied (Fig. 2c, d). It is challenging to create colors with separate reflectance peaks within the wavelength sensitivities of non-adjacent color cones because the peaks must be highly saturated to avoid stimulating neighboring cones[2]. However, this idea does not explain why there are far more true purple (blue + red) than UV/V + green plumage colors. Notably, birds particularly fail to fill the more UV/V regions (those closer to the UV/V vertex) of UV/V + green color space, which might indicate that it is difficult to create spectra with *uv/v* wavelength peaks higher than those in the *m* wavelengths.

The differences between our methods and those of Stoddard and Prum[2] likely contribute in part to the larger gamut size when comparing species data but not overall data. While the number of species included in our study was comparable to that of Stoddard and Prum[2] (114 vs 111 species, respectively), we measured almost twice as many plumage patches as they did (+1600 vs. 965

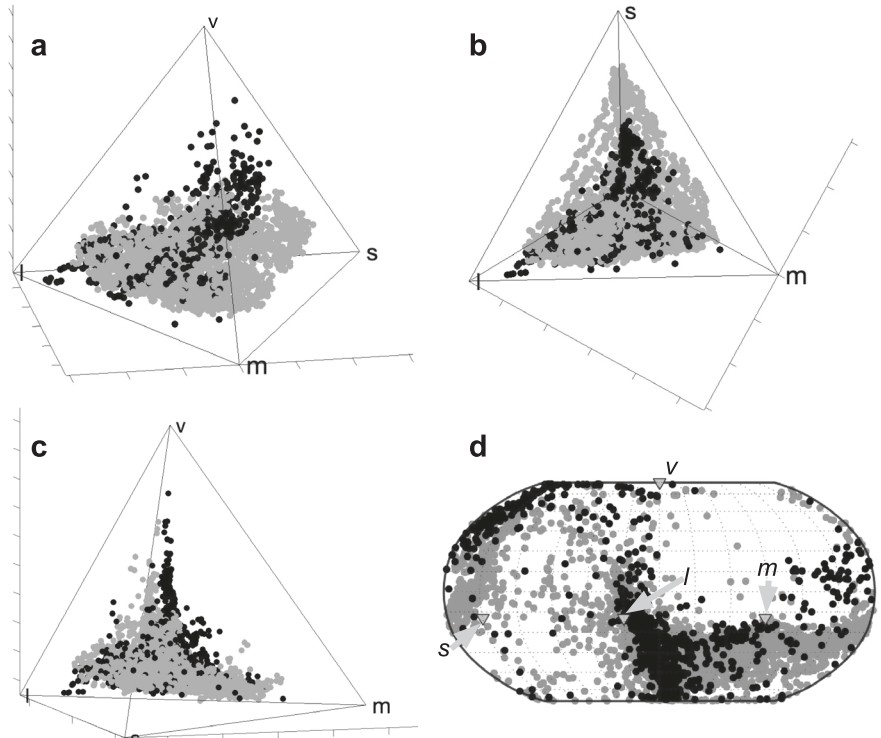

**Fig. 2 The contribution of hummingbird spectra to the known avian color gamut. a–c** The distribution of color points of our hummingbird data (gray) and Stoddard and Prum's (2011) data for all birds (black) in an avian VS color space viewed from different directions: **a** standard angle, **b** a top-down view displaying the expansion of hummingbird plumage colors in s-, m-, and l- dimensions, and **c** a view toward the l-m edge from slightly below displaying the expansion of hummingbird colors toward the s-v-l face of the color space, and the paucity of color points in the m+ uv/v region of the color space. **d** Corresponding Robinson projection of color point hues showing the expansion of hummingbird colors in terms of hues, especially in the s−, s+ m−, s+l −, and s+l+uv/v− regions.

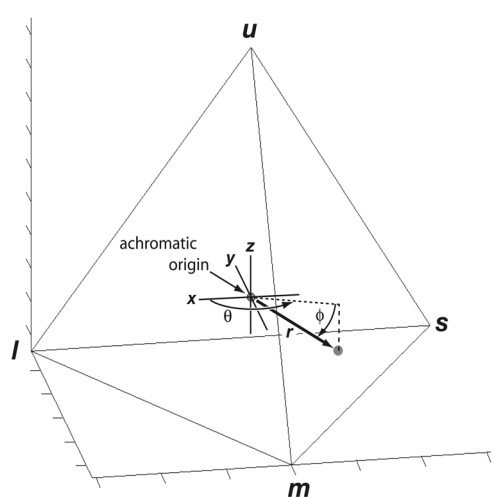

**Fig. 3 Tetrachromatic avian color space.** The avian perception of a reflectance spectrum is represented as a point in color space determined by the relative stimulation of the ultraviolet or violet (uv/v), blue (s), green (m), and red (l) retinal cones. The center of the tetrahedron is the achromatic origin, which represents equal cone stimulation– either white, black, or gray. Each color point is defined by the angles θ and φ, which define the hue, and the distance r, which defines the chroma or saturation (reproduced from Stoddard and Prum 2008).

patches). To prevent erroneous distortion to iridescent colors we did not average the three measurements per patch. Both studies measured six standard patches for all species and additional patches if necessary to capture other plumage color variation. The larger number of plumage patches we measured reflects how color diverse hummingbird plumages are. Our methods preserved the natural variation in hue due to iridescence and avoided the distorted flattening caused by averaging highly saturated peaks with slightly different peak hues. Although our methods are biased toward increasing variation, they are necessary to accurately capture the phenomenon of iridescent hummingbird coloration.

There are multiple reasons why the hummingbird color gamut is so diverse. The size of the hummingbird color gamut, like the achieved color gamut of any clade, constitutes a combination of the history of selection on color function, the clade's evolved capacities for color production, the age of the clade, and the number of species. Hummingbirds excel at all these criteria. The 336 species of extant hummingbirds have radiated rapidly over the last 22 million years[19]. Hummingbird plumage color diversity has evolved through a long history of persistent sexual and social selection on plumage coloration. Hummingbirds have polygynous breeding systems characterized by female only parental care, female mate choice, and often elaborate male courtship displays. Intersexual selection in hummingbirds has contributed to elaborate radiation in brilliant plumage coloration as well as vocalizations and non-vocal feather sounds[14,16,20]. Hummingbird plumage color evolution rates have even been shown to positively correlate with hummingbird speciation rates[14]. Furthermore, in some species, brilliant monomorphic plumage ornaments apparently function in aggressive, intra- and interspecific defense of floral resources[21] and appear to be associated with socioecological features related to resource competition[19]. Our finding that crown and throat patches, which flash brilliantly when the head of the bird is oriented toward the observer, are more diverse in coloration than

**Table 2 Comparison of the 15 largest species color gamuts from this study and Stoddard and Prum (2011).**

| Species | # of patches | Measurements per patch | Color System | Volume | % of color space occupied |
|---|---|---|---|---|---|
| Boissonneaua jardini | 18 | 3 | VS | 2.98E-02 | 13.8 |
| Panterpe insignis insignis | 28 | 3 | VS | 2.15E-02 | 9.93 |
| Chrysuronia oenone oenone | 20 | 3 | VS | 1.42E-02 | 6.56 |
| Amazilia rosenbergi | 16 | 3 | VS | 1.31E-02 | 6.05 |
| Aglaiocercus coelestis coelestis | 13 | 3 | VS | 1.30E-02 | 6.00 |
| Heliactin bilophus | 16 | 3 | VS | 1.13E-02 | 5.22 |
| Oreonympha nobilis nobilis | 19 | 3 | VS | 1.11E-02 | 5.13 |
| Coeligena iris | 16 | 3 | VS | 1.10E-02 | 5.08 |
| Augastes scutatus scutatus | 16 | 3 | VS | 1.09E-02 | 5.03 |
| Juliamyia julie panamensis | 13 | 3 | VS | 1.05E-02 | 4.85 |
| Charmosyna papou- S&P | 13 | 1* | UVS | 1.01E-02 | 4.67 |
| Tangara chilensis- S&P | 9 | 1* | UVS | 9.17E-03 | 4.24 |
| Amazonia albifrons- S&P | 13 | 1* | UVS | 6.51E-03 | 3.01 |
| Trichoglossus rubritorquis- S&P | 10 | 1* | UVS | 3.95E-03 | 1.82 |
| Erythura gouldiae- S&P | 8 | 1* | UVS | 3.80E-03 | 1.76 |

Ten hummingbird species have larger color gamuts than the five most color diverse species found by Stoddard and Prum[2]. *indicates that three measurements of each were averaged together. Note that we took more measurements than Stoddard and Prum[2] and did not average them (in order to preserve within patch variation).

**Table 3 Summary of VS color measurements for each patch type.**

| Patch Type | # of measurements | Volume | % of Color Space | % of Avian Gamut | Average Color span | Max span | Average Hue disp | Max Hue disp | Average chroma |
|---|---|---|---|---|---|---|---|---|---|
| **Throat** | 747 | 6.72E-02 | 31.0 | 76.6 | 3.04E-01 | 8.09E-01 | 1.27E+00 | 3.13E+00 | 2.40E-01 |
| **Crown** | 463 | 5.01E-02 | 23.1 | 57.1 | 3.20E-01 | 8.19E-01 | 1.30E+00 | 3.13E+00 | 2.62E-01 |
| **Back** | 664 | 3.61E-02 | 16.7 | 41.2 | 1.85E-01 | 7.34E-01 | 7.55E-01 | 3.13E+00 | 2.37E-01 |
| **Belly** | 730 | 3.60E-02 | 16.6 | 41.0 | 2.05E-01 | 7.32E-01 | 8.70E-01 | 3.13E+00 | 2.05E-01 |
| **Tail** | 744 | 3.29E-02 | 15.2 | 37.5 | 1.85E-01 | 7.63E-01 | 1.30E+00 | 3.14E+00 | 1.43E-01 |
| Rump | 275 | 2.02E-02 | 9.32 | 23.0 | 1.94E-01 | 6.90E-01 | 9.05E-01 | 3.13E+00 | 2.10E-01 |
| Nape | 248 | 1.65E-02 | 7.62 | 18.8 | 1.94E-02 | 6.37E-01 | 8.52E-01 | 2.93E+00 | 2.25E-01 |
| Cheek | 84 | 1.09E-02 | 5.05 | 12.4 | 1.69E-01 | 6.39E-01 | 8.87E-01 | 2.93E+00 | 1.42E-01 |
| Undertail coverts | 384 | 7.90E-03 | 3.65 | 9.01 | 1.31E-01 | 4.83E-01 | 7.79E-01 | 3.14E+00 | 1.30E-01 |
| **Wing** | 583 | 5.66E-03 | 2.62 | 6.45 | 1.17E-01 | 4.19E-01 | 7.77E-01 | 3.13E+00 | 1.18E-01 |
| Leg | 78 | 4.35E-05 | 0.02 | 0.05 | 4.70E-02 | 3.09E-01 | 2.22E-01 | 1.43E+00 | 6.80E-02 |

Patches are listed in order from largest gamut size (i.e., volume) to smallest gamut size. Standard patches are in bold.

other plumage regions highlights the role of plumage coloration in direct inter-individual communication and social interactions.

The mechanistic properties of hummingbird barbule structural color further explain the exceptional diversity of hummingbird plumage coloration. Hummingbird barbule structural coloration is among the most complex plumage coloration mechanisms, comprised of stacks of hollow, air-filled melanosomes, surrounded by a thin superficial, solid keratin cortex as well as sometimes superficial, miniature melanin platelets which lie just beneath this cortex[9–13]. Complex nanostructures allow for independent tuning of multiple components, and, hence, greater achievable color diversity[12,18,22]. Barbule structural color permits the production of any peak-reflected wavelength by varying the thickness of melanosome arrays, which can produce a diversity of single-peak spectra-hues, such as the unusual diversity of greens, blues, and blue + greens seen in hummingbirds (Fig. 2b). Hummingbird melanosomes are among the most unusual in birds in being both disc-shaped and air-filled[9–13,23]. The air in the center of hummingbird melanosomes approaches the maximum possible biological difference in refractive index (air = 1.0, melanin = ~1.7), which results in the efficient production of brilliant colors with the fewest layers of melanosomes, such that resulting spectra are narrow and near saturation[13,24]. Such spectra can

thereby create colors that extend further in color space (Fig. 2a–c).

Barbule structural color also allows for the production of plumage spectra with multiple saturated peaks, creating saturated color combinations that are not as commonly produced via other plumage coloration mechanisms. However, researchers have yet to identify exactly how hummingbird multipeak spectra are produced[12,13], emphasizing the need for further analyses of the optics of hummingbird feathers. Many hummingbird melanosome arrays are non-ideal– i.e., the products of the thicknesses and refractive indices of the melanin and air cavity layers are not equal[25]. Non-ideal thin films can create more highly saturated, pure tone colors of the primary peak while also introducing additional, harmonic spectral peaks at shorter wavelengths[25], which allows for complex reflectance spectra with multiple bright peaks within the avian visible spectrum. Also, melanosome arrays with a large average layer thickness (>~300 nm) can create colors with fundamental interference peaks in the infrared and multiple, harmonic peaks in the avian visible range (300–700 nm). The presence of minute, superficial melanin platelets below the cortex in hummingbird barbules is also correlated with secondary, lower wavelength reflectance peaks, but the precise optical mechanism remains to be established[12]. These different nanostructural elements all contribute to distinctive multipeak reflectance spectra

that can stimulate non-adjacent color cone combinations, which Stoddard and Prum[2] identified as particularly difficult to accomplish: UV/V-purple ($uv/v + s + l$ wavelengths; *Schistes*

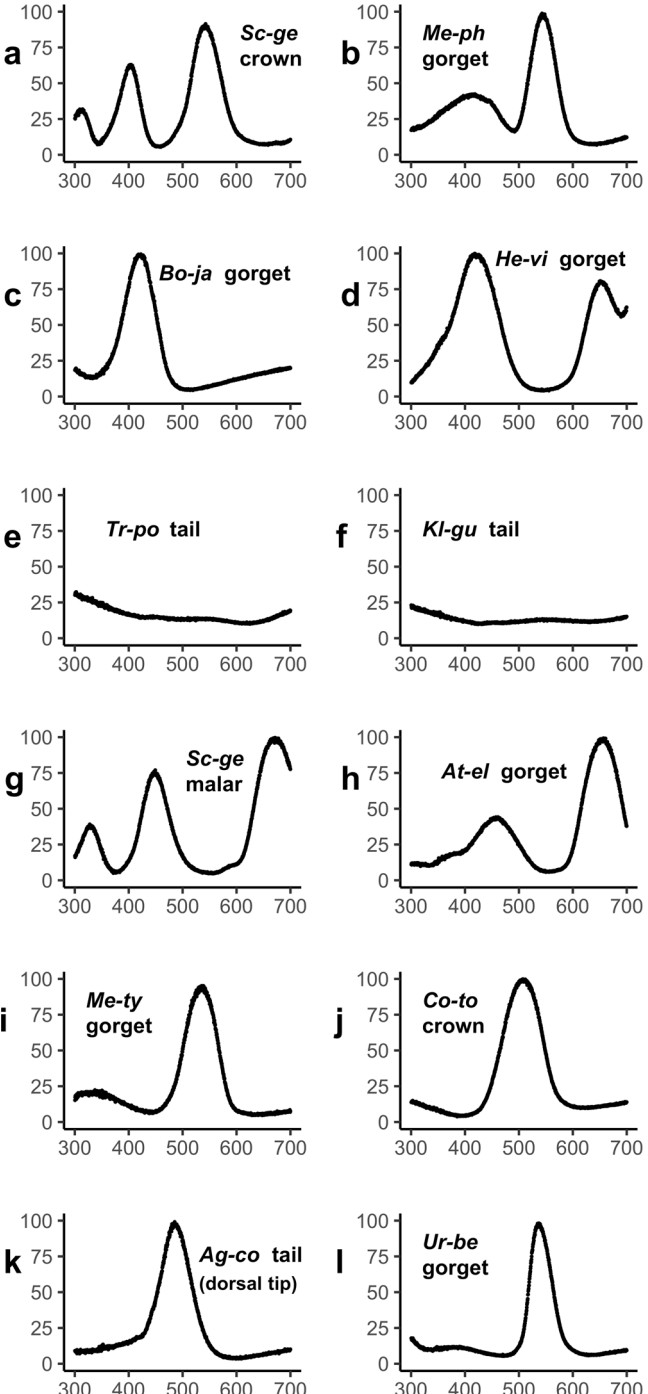

**Fig. 4 Examples of unusual and distinctive hummingbird reflectance spectra. a, b** saturated green + UV reflectance, **c, d** UV saturated reflectance, **e, f** low reflectance spectra with unusual UV component, **g, h** true purple reflectance (i.e., $s + l$), **i, j** saturated blue + green ($m + l$) reflectance, **k** highly saturated blue, and **l** highly saturated green (right) spectra. Spectra are labeled with the first two letters of the genus and species names and the patch: **a, g** *Schistes geoffroyi*, **b** *Metallura phoebe*, **c** *Boissonneaua jardini*, **d** *Heliangelus viola*, **e** *Trochilus polytmus*, **f** *Klais guimeti*, **h** *Atthis ellioti ellioti*, **i** *Metallura tyrianthina smaragdinicollis*, **j** *Coeligena torquata fulgidigula*, **k** *Aglaiocercus coelestis coelestis*, and **l** *Urosticte benjamini*. Reflectance spectra are in arbitrary units (See Methods).

*geoffroyi* cheek, Fig. 4g); true purple ($s + l$ wavelengths; *Atthis ellioti* gorget, Fig. 4h); UV/V-green ($uv/v + m$; *Schistes geoffroyi* crown, Fig. 4a); and UV/V-red ($uv/v + l$; *Heliangelus viola*, Fig. 4b). With multipeak spectra the potential for creating new and different colors is greatly expanded, allowing for a more versatile evolution of novel colors.

Unexpectedly, the hummingbird plumage color gamut is larger in volume when modeled with the VS-type (34.2%) than with the UVS-type (29.6%) visual system. This apparently unique result contrasts notably with both Stoddard and Prum's[2] and our revised estimate of the color gamut of all birds combined– VS gamut = 40.5%; UVS gamut = 47.3%. Multiple previous analyses have shown that the UVS cone-type visual system does a more efficient job of discriminating the colors of natural objects because of the broader separation between the peak spectral sensitivities of the *uv* and *s* (blue) cone types[2,26,27]. Because the UVS-type visual system produces an even greater increase in color volume for a diverse plant color data set over the VS-type visual system, Stoddard and Prum[2] rejected the hypothesis that the UVS-type visual system had specifically evolved to expand the diversity of avian color stimuli.

However, our observations that the hummingbird plumage gamut is substantially *greater* in volume with the VS-visual system than with the more efficient UVS-visual system strongly suggests another hypothesis: Hummingbird plumage may have specifically evolved to be more diverse within the hummingbird VS-type color visual system via selection for highly saturated plumage colors. Given diversity in hue, the way to achieve greater color gamut volume, i.e., greater plumage color diversity, is through highly chromatic color vectors that extend toward the limits of the color space. The two visual systems map variation in wavelength to different maximum potential chroma—i.e., wavelengths with color vectors that extend toward the edges, faces, and vertices of the tetrahedron[6]. Color vectors that extend towards the vertices, i.e., plumage that best corresponds to a singular cone type's peak sensitivity, have the highest maximum potential chroma because vertices are the regions furthest away from the tetrahedron's center. Thus, hummingbird plumages may have specifically evolved to have maximum chroma within their own VS-visual system via peaks that correspond most closely to the peak sensitivities of the VS- rather than the UVS-visual system. For example, when comparing the UVS and VS plumage color gamuts for hummingbirds, it is notable that hummingbird coloration extends much further into the UV/V regions of color space for the VS-visual system (Supplementary Fig. 2). While in the VS system these color points map toward the *v* vertex, in the UVS-visual system they map towards the *uv-s* edge and the *uv-s-l* face. Such color vectors that contribute to expanded color volume of the VS gamut could have evolved by sexual or social selection for highly saturated plumage colors that are near in hue to the specific sensitivity peaks of hummingbird receptor cone types. Such selection could note preferences within some hummingbird species for hues with maximally possible chroma, not merely for maximal chroma of a given hue.

Hummingbirds have tetrachromatic color vision with substantial sensitivity in the near ultraviolet[28,29]. Recently, Stoddard et al.[30] used a series of elegant experiments with hummingbird feeders and LED lights to demonstrate for the first time that hummingbirds can distinguish non-spectral colors distributed throughout the tetrachromatic color space. However, the presence of this remarkably proficient four-color vision in hummingbirds poses an interesting evolutionary conundrum. Recent phylogenetic analyses have established that hummingbirds and swifts are phylogenetically embedded *within* the nocturnal caprimulgiforms[31,32]. The most parsimonious hypothesis is that the immediate ancestors of swifts and hummingbirds were extensively nocturnal for approximately 8

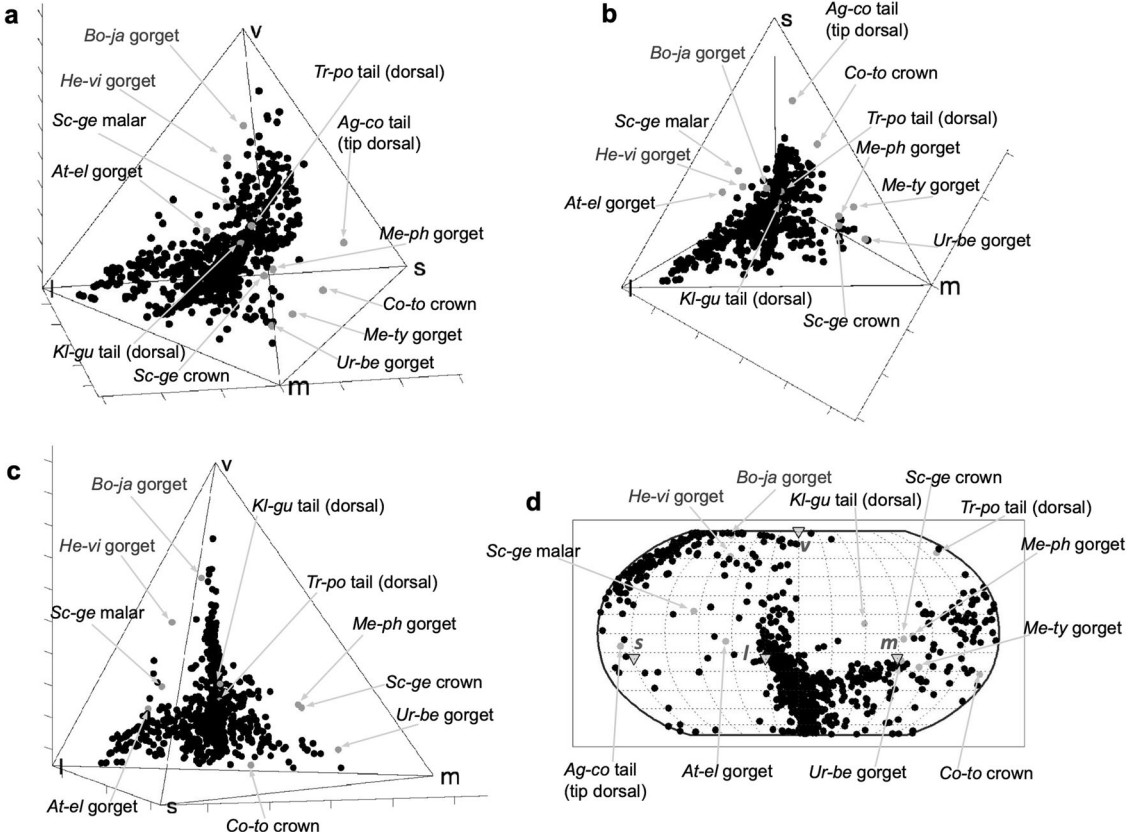

**Fig. 5 Distribution of unusual hummingbird plumage spectra in color space.** Unusual hummingbird plumage spectra (gray) from Fig. 4 with the total avian color gamut (black) from Stoddard and Prum (2011) **a–c** plotted in avian VS color space and viewed from three angles: **a** standard view, **b** a top-down view, and **c** a view toward the *l-m* edge from slightly below. **d** Robinson projection of the hues of the same color points. All spectra are labeled with the first two letters of the genus and species: *Ag-co– Aglaiocercus coelestis coelestis, At-el– Atthis ellioti ellioti, Bo-ja– Boissonneaua jardini, Co-to– Coeligena torquata fulgidigula, He-vi– Heliangelus viola, Kl-gu– Klais guimeti, Me-ph– Metallura phoebe, Me-ty– Metallura tyrianthina smaragdinicollis, Sc-ge– Schistes geoffroyi, Tr-po– Trochilus polytmus*, and *Ur-be–Urosticte benjamini*.

million years before they re-evolved diurnal ecology and behavior[31]. Given that an evolutionary history of nocturnality can lead to the degradation or loss of opsin genes[33,34], it should be a high priority to establish what effect that ancestral nocturnality may have had on the molecular physiology and anatomy of the hummingbird color visual system.

Our attempt to document the color diversity of an avian family has revealed that current estimates of the total avian color gamut are likely inaccurately low. Similar studies sampling from other color-diverse families, such as sunbirds (Nectariniidae), parrots (Psittacidae), tanagers (Thraupidae), birds of paradise (Paradiseidae), manakins (Pipridae), and starlings (Sturnidae), most of which have already been studied for their plumage coloration[35–39], would help us obtain a better estimate of the true avian color gamut.

## Methods

**Plumage color measurement.** More than 5000 reflectance spectra were collected from over 1600 plumage patches from 114 different species of hummingbirds (Trochilidae) from study skin specimens from the Yale Peabody Museum (YPM) and the American Museum of Natural History (AMNH) (species and catalog numbers are listed in Supplementary Data 1). The sample included approximately 33% of extant hummingbird species (and 60% of genera). To maximize the estimated hummingbird plumage color gamut, species were selected for unique colors or diverse plumages (as seen by human observers). To represent the diversity of the family, several species were included from each major clade as defined by McGuire et al.[19] regardless of coloration. Mostly male specimens were used as they are usually the most colorful sex.

Following Stoddard and Prum[2], reflectance spectra were measured from six standardized patches from all specimens: crown, back, tail, wing, belly, and throat.

Additional patches were measured if they were distinct to the human eye and large enough to measure reliably. Each plumage patch was measured from a different position three times. Multiple measurements were not averaged to prevent flattening of highly saturated peaks with slightly different hues. If a patch showed a gradient in color, measurements were taken at the ends and center of the gradient.

Reflectance spectra were measured with a S2000 Ocean Optics spectrometer and a bifurcated fiber with an Ocean Optics DH-2000-BAL deuterium–halogen light source (Ocean Optics, Dunedin, FL) in a dark room with an integration time of 100 ms. We did not use a metal block to hold the reflectance probe because it can be difficult to accurately measure plumage reflectance peaks from small iridescent patches at a normal angle of incidence to the plumage surface. Rather, we used a Keysight 3D Probe Positioner (N2787A, KEYSIGHT, Santa Rosa, CA) to hold the optical fiber stable at the appropriate angle of incidence to maximize peak reflectance and saturation while maintaining peak reflectance below 100%.

Each color patch was scored with a presumed color production mechanism, including barbule structural color, melanin (further classified as either eumelanin or phaeomelanin), or white (unpigmented). Color production mechanisms were inferred based on previous literature[40], visual appearance, and the shape of the reflectance spectra. Any color that showed barbule structural color as well as melanin was categorized as a barbule structural color. Structural black colors[41] were excluded from analyses.

**Color space analyses.** We modeled avian perception of color using a tetrahedral color space[2,6] (Fig. 3) to provide a quantitative representation of avian sensory experience. We used the shareware computer program TETRACOLORSPACE 1.0 (TCS) for MATLAB[2,6]. The way TCS works was described in Stoddard and Prum (2011)[2]:

"The idealized stimulus, $Q_I$, of each color cone-type was estimated by the reflectance spectrum of a plumage patch:

$$Q_1 = \int_{300}^{700} R(\lambda)C_r(\lambda)d\lambda \tag{1}$$

where $R(\lambda)$ is the reflectance spectrum of the plumage patch and $C_r(\lambda)$ is

the spectral sensitivity function of each cone-type $r$. $R(\lambda)$ and $C_r(\lambda)$ functions were normalized to have integrals of 1. We assumed a standard constant illumination across all visible wavelengths. For each plumage color, the idealized stimulation values of the 4 color cones—$Q_I$—were normalized to sum to one, yielding relative {$uv/v\ s\ m\ l$} values.

The {$uv/v\ s\ m\ l$} values of each reflectance spectrum were converted to a color point with spherical coordinates θ, φ, and r, which define a color vector in the tetrahedral color space... This tetrahedral geometry places the achromatic point of equal cone stimulation—white, black, or gray—at the origin and the $uv/v$ vertex along the vertical $z$-axis… Each color has a hue and saturation. Hue is defined as the direction of the color vector, given by the angles θ and φ, which are analogous to the longitude and latitude, respectively… Saturation, or chroma, is given by the magnitude of $r$, or its distance from the achromatic origin. Because the color space is a tetrahedron and not a sphere, different hues vary in their potential maximum chroma, or $r_{max}$ (Stoddard and Prum 2008)[6].”

We estimated the total gamut of hummingbird plumage coloration, and gamuts of specific coloration mechanisms, species, and plumage patches by calculating the volume of color space occupied by the minimum convex polygon containing all relevant color points. We used Robinson projections to view the distribution of hues independent of saturation[2,5,6].

For the entire hummingbird color sample, each plumage patch type, and each species, we calculated a plumage color volume, the % volume occupied of the total color space, the average and maximum color span (i.e., linear distance between color points), and the average and maximum hue disparity (i.e., difference in color vector angle between color vectors). We calculated color volumes in terms of the % of the achieved color gamut of all birds, both as estimated by Stoddard and Prum[2] and by combining their data with our expanded hummingbird data. Lastly, we created tetrahedral plots and Robinson projections for all hummingbirds, for each hummingbird species, for each patch type, for each color production mechanism type, and for our hummingbirds combined with Stoddard and Prum[2]’s data. Following the methods of Stoddard and Prum[2], we also calculated color span, hue disparity, and chroma metrics. Hummingbirds appear to have a violet-sensitive (VS) type color visual system[30,42] (but see[29,43,44]); therefore our results are reported primarily using VS visual system, but all analyses were conducted using both VS and ultraviolet sensitive (UVS) visual system settings for comparison.

The use of convex hull volumes has been criticized as containing unoccupied spaces[45] and being sensitive to outliers[46]. However, we use the convex hull rather than other recently proposed alternatives, such as Delhey 2015’s color space grid[45] or Gruson 2020’s α-shapes[46], because the convex hull appropriately quantifies the perceivable signal diversity and total contrast achieved by a clade within a sensory space. As proposed by Stoddard and Prum[6], the color gamut is a macroevolutionary concept, capturing the achieved sensory diversity of a species or clade. The sensory perception of the organisms themselves, and the capacity for contrast among those perceptions, are central to the intellectual goal of quantifying a color gamut. Because each of the edges of a convex hull is a potentially observable color contrast between color points, each makes a legitimate contribution to the estimate of the volume of color space occupied. The α-shapes method of estimating color space occupancy[46] has the additional problem that it cannot establish a standardized α-value for comparing the gamuts based on different heterogeneous samples. Thus, it is not appropriate to the goals of this study.

**Statistics and reproducibility**. Using the Keysight 3D Probe Positioner (N2787A, KEYSIGHT, Santa Rosa, CA) to help collect plumage color reflectance spectra allowed us to preserve some within patch color variation while also increasing repeatability of spectra from iridescent structural colored patches because it allowed us to find the appropriate angle that produced the most saturated spectra: The most saturated spectra exhibit normal incidence to the laminar nanostructures in the barbules, which may not be normal to the plane of the plumage surface. As a consequence, however, the distance of the optical fiber from the plumage surface could not be standardized, so reflectance spectra are reported in arbitrary units rather than % reflectance. This has no impact on color space analysis, which ignores spectral brightness.

To see how well our sample estimates the total hummingbird plumage color gamut, we resampled using a rarefaction scheme to see how the color gamut increased with the number of species via associated error terms. We resampled datasets of 60, 70, 80, 90, 100, and 110 species without replacement 10 times each (Supplementary Data 1) and calculated the average, standard deviation, and coefficient of variation in plumage color gamut size.

**Reporting summary**. Further information on research design is available in the Nature Research Reporting Summary linked to this article.

## Data availability
The data depicted by the figures and tables are available on Dryad (doi:10.5061/dryad.1c59zw3xn)[47]. Individual spectrum TXT files for each measurement are also available from the corresponding authors upon reasonable request. Skin specimens were

from the Yale Peabody Museum and the American Museum of Natural History (Catalog numbers located in Supplementary Data 1). The data from Stoddard and Prum 2011[2] are available at https://doi.org/10.1093/beheco/arr088).

## Code availability
TetraColorSpace software available online at https://www.marycstoddard.com/software. An R-script used to combine raw spectra data (individual TXT files) to CSV files with the data for each species is also available on Dryad (https://doi.org/10.5061/dryad.1c59zw3xn)[47].

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

## Acknowledgements

This work was completed in fulfillment of the undergraduate thesis requirement for E&EB at Yale University by GXV. We thank the Yale Peabody Museum of Natural History (YPM), and the American Museum of Natural History (AMNH) for permission to measure specimens in their care, especially Kristof Zyskowski and Bentley Bird. David Heiser and Dave Evans provided advice to GXV on funding. Glenn Bartley, Wilmer Quiceno, and John Cahill kindly gave permission for us to reproduce their lovely photographs (Fig. 1). The research was supported by a Yale Peabody Museum Summer Internship Fellowship and the Yale Berkeley College Max and Reba E. Richter Scholarship and Robert Berlin Fellowship to G.X.V., and the Yale William Robertson Coe Fund to R.O.P.

## Author contributions

G.X.V. and R.O.P. formulated the questions and wrote the paper; G.X.V. collected and analyzed the data; K.G. created the R-script necessary to analyze the data; R.O.P. supervised research.

## Competing interests

The authors declare no competing interests.
