## [Peer Review File · Communications Biology]

Reviewers' comments:

Reviewer #1 (Remarks to the Author):

In this manuscript the authors use a large collection of reflectance spectra obtained from hummingbirds (family Trochilidae) and compare their colour gamut (the range of colours they can produce) with that of all birds based on the data in Stoddard & Prum 2011 Behav Ecol <https://doi.org/10.1093/beheco/arr088>.

They conclude that the hummingbirds occupy a larger portion of the colour space of birds than all other species combined (note that not all bird species were measured, the estimate of the total gamut of colours in birds was based on a sample of 111 species). The high colour diversity of hummingbirds is probably due to the fact that one of their main mechanisms of colour production is barbule structural colours, which can in theory produce nearly any hue and the fact that sexual selection is strong in this family (all or nearly all species have female-only parental care with polygynous social mating system).

This is a very interesting manuscript and while I think that the main conclusion is going to be supported over time, I have a few concerns that need to be addressed as detailed below:

Main concerns

1) Authors should be a bit more circumspect when calling hummingbirds the most colour-diverse family. The fact that sampling a fraction of hummingbirds species expands the colour gamut of all birds highlights that the latter estimate was not an accurate representation of the extent of colour diversity of birds. This means that further sampling may reveal other highly colour-diverse families. While I feel that it is likely to be true that hummingbirds shall be the most colour-diverse family, similar in-depth comparisons with other intensively sampled families were not attempted (eg sunbirds, parrots, tanagers). Hence, I suggest to qualify these statements throughout and potentially outline which other families could be contenders that should be assessed in a similar manner. This might prompt further exploration by other researchers (some of whom may already have the relevant data).

2) The use of convex hull volume to assess colour gamut has been criticised and alternative methods have been suggested (e.g. Delhey 2015 Sci Rep <https://doi.org/10.1038/srep18514>; Gruson 2020 Methods Ecol Evol <https://doi.org/10.1111/2041-210X.13398>). It would be good to make use of these approaches (in particular the one outlined in Gruson 2020) as alternative ways to quantify total colour space occupied. The main problem of convex hull volume is that it mostly contains empty space and is highly sensitive to outliers.

3) Similarly, it would be good to carry out resampling procedures to assess how much specific species contribute to the values reported. This may allow to compute confidence intervals around the values reported.

4) As far as I understand the methods used to collect reflectance spectra differ between this manuscript and Stoddard & Prum 2011. I think that it would be good to add a bit of discussion on the potential for this to affect conclusions.

Specific minor comments (Ln = line number)

Ln 39 complement

Ln 105 how were melanin patches defined?

Ln 131 as currently known

Ln 161 why 'ultra-red'? do you mean ultraviolet-red?

Ln 174 true purple?

Ln 188 densely, how did you define this?

Ln 193 should a statement like this not be accompanied by some estimate of evolutionary rates? A specific clade may be able to evolve a large colour gamut simply because it had a long time to do so.

Ln 194-196 I think that some further elaboration as to what mechanism determines the difficulty of evolving/producing such colours would be good here.

Ln 197-200 the argument being that uv+green colours are not very conspicuous (because they match the green of leaves) nor very cryptic (because they have the UV component that is not so marked in leaves)? Because otherwise you could assume that green colours could evolve because they are cryptic in a vegetated environment. Many green colours that are meant to be cryptic do have a UV component (e.g. parrot green). It might be interesting to compare how well/bad these green colours match the spectrum of green leaves.

Ln 205-206 +time available.

Ln 254 revise writing

Ln 267-271 do you think that this would be an example of matching between sensory system and signal as predicted by the sensory drive theory? Perhaps some mention would be required, plus some further ideas on how to test this. Previous studies in birds are a bit equivocal.

Ln 288-290 some further discussion of the ideas outlined in Wilkins & Osorio 2019 J exp Biol <https://doi.org/10.1242/jeb.204487> would be good here.

Ln 311 as assessed by human observers.

Reviewer #2 (Remarks to the Author):

The purpose of this research manuscript was to explore the plumage color gamut of the hummingbird family (Trochilidae) based on 114 species of hummingbirds. This research project builds on the work by Stoddard and Prum (2011) that had previously established the avian color gamut for 111 species of birds representing the known diversity of avian color production mechanisms. Stoddard and Prum (2011) observed structural colors made the largest contribution to the avian color gamut and predicted that greater sampling of structural colors would further expand the avian color gamut. This manuscript specifically tests this prediction with hummingbirds, which are known for their diverse and complex barbule structural color. In addition to this, the manuscript also further expands on the work of Stoddard and Prum by examining what constraints avian coloration mechanisms have on the size of the avian color gamut.

The authors found that the structural colors of hummingbirds occupy 34.2% of the color space perceived by birds and demonstrate that the hummingbird structural color gamut expands on the previously established space avian color gamut determined by Stoddard and Prum (2011). The increases to the avian color gamut are all due to contributions of hummingbird structural colors that were not found in the taxa used in the original study. These results provide further support for the hypothesis that barbule structures impose the fewest constraints on the diversification of plumage coloration. The authors also determined that the size of the gamut volumes associated with specific plumage patches were related to differences in their life history roles.

This study is an important contribution to expanding our knowledge of the relationship between the avian visual system and the coloration expressed by species of birds. The original study by Stoddard and Prum (2011) provided a preliminary overview of the avian color gamut using a relatively small sample of species (compared to the total number of known avian taxa) representing a diverse range of taxa from several different orders. The current study is the next

step in getting a closer look at the color gamut within a family of closely related species that exhibit diverse coloration now that the method and tools have been established. Of all of the avian species, hummingbirds would be the family to further test the contributions of structural colors to the avian color gamut. I believe the sample size and the choice of taxa used was appropriate for the hypotheses tested in this study and the data collection was thorough, with repeated measures of each individual for each of the six standardized patches. The methods outlining the spectra measures are detailed and well supported in the literature. The authors do an excellent job explaining the color gamut of the hummingbird structural colors in terms of their contributions to the total avian color gamut and also look closely at the color gamuts of specific species that expanded the volume of the avian color gamut. The figures are informative and easy to understand. I also believe the discussion is well-written. The explanations for the potential physiological and structural mechanisms behind why hummingbirds are so colorful are detailed and support the results reported in the study. My only issue is that I feel that the discussion ends abruptly without any discussion of next steps, and particularly whether or not there is a need to investigate the plumage coloration and mechanisms of color in other families of birds with these fine-scale analyses to investigate the avian plumage color gamut further.

Below, I list specific comments I have for the authors:

Lines 87-88: You say in the methods that 114 species from 68 genera represent 33% of extant hummingbird species, but I think this needs to be included in the main text here along with whether or not you have at least one representative of each of the current known genera. I think this is important because one of the main goals of the project is to expand on the sampling done for structural colors in the original Stoddard & Prum (2011) data set.

Line 109: It would be useful to know how many species of hummingbirds were included in these previous analyses of avian color diversity so that it is clearer how this study has improved upon the sampling in these previous studies.

Line 131 & Line 146: I think this should be reworded to say so that it states that hummingbirds are the most diversely colored family of birds in the world based on the current data set. One aspect that is not clear in the paper is if this type of analysis has been done for any other bird family other than hummingbirds. I realize that to the human eye hummingbird coloration may look the most diverse, but this claim in line 131 is hard to justify without far more data on other families of birds to match the amount of data that we now have on hummingbirds. In line 147 you do state that hummingbirds contain nine of the most diversely colored species of known birds, but this needs to be clarified in terms of describing families.

Line 190: You need a reference here demonstrating that throat and crown plumage patches play important roles in hummingbird communication

Line 257: I don't understand this heading what do you mean by larger vs. hummingbird color gamut size? Based on the text in the section below it seems as though you are comparing the results in different visual systems and the subtitle should reflect that.

Lines 292-303: The last paragraph here ends the paper abruptly. You imply that you believe having an ancestor that was nocturnal may contribute to the diversity of structural coloration observed in hummingbirds, but I'm not clear why you predict this would be the case. I would like to see the end of this paper in a few sentences focus on next steps; particularly whether or not there is a need to investigate the plumage coloration and mechanisms of color in other families of birds with these fine-scale analyses to investigate the avian plumage color gamut further. This is an impressive amount of work, but the question is does it need to be done in other families or is this a good stopping point for our understanding of the avian color gamut?

Line 312: Is clade the same as your 68 genera included in line 88? Does this mean you have a representative species from each currently known genera of hummingbirds? I think this needs to be clarified.

References:

Stoddard, M. C. & Prum, R. O. How colorful are birds? Evolution of the avian plumage color gamut. *Behavioral Ecology* 22, 1042-1052 (2011).

Reviewer #3 (Remarks to the Author):

I have serious concerns about this work, both conceptual and methodological. First, from the title it may be inferred that the color gamuts of all species of birds, or a significant proportion of species, have been investigated and compared with those of hummingbirds. However, the only reference study is Stoddard and Prum (2011), who measured the amplitude of the color gamut of 111 species of birds. Far from being a significant representation of all extant birds. The conclusion of the present study that hummingbirds have the greatest plumage color diversity of all birds and that this represents a considerable expansion of the avian color gamut as previously estimated, thus, is not fully justified. It may be an expansion of the color gamut provided by the previous study of the team (Stoddard and Prum), but not of the known avian color gamut as stated.

My second concern refers to how color is considered and measured here. Color is not only a neural process output in the receivers. Color is, firstly, a physical property of the objects. It is thus an intrinsic property of the objects, not only something that occurs in the brain of the animals that perceive it. I recommend the authors to read the works by Francisco J. Varela on this topic. As a consequence, color should not be measured by tetrahedral or any color space models. To be objective and obtain results that are comparable among taxa (how could we compare the results of this article with the color gamut of mammals, for example?), color diversity should be quantified by studying variability in the physical phenomenon, i.e. the electronic spectra.

It is also surprising that the measurement of color in hummingbirds was done in a standardized way considering the same plumage patches in all species, but then, if there were other patches with a color that was distinctive to the eyes of the authors, these were also measured. This is equivalent to measure the colors that the authors were perceiving on birds. The amplitude of the color gamut is therefore constrained by the human perception of distinct colors, as the colors that were measured depended on what the authors perceived.

Lastly, the discussion is exceedingly long and speculative. For example, there are many mentions to variability in morphological features of barbules causing differences in color, but no morphological measures are provided in this study.

We appreciate the time and effort spent on the review of this manuscript. There were many insightful comments that we feel have improved it. In the following, we will address the comments one at a time. We quote the original comments verbatim in italics, and our responses are in non-italics.

Reviewers' comments:

Reviewer #1 (*Remarks to the Author*):

In this manuscript the authors use a large collection of reflectance spectra obtained from hummingbirds (family Trochilidae) and compare their colour gamut (the range of colours they can produce) with that of all birds based on the data in Stoddard & Prum 2011 Behav Ecol <https://doi.org/10.1093/beheco/arr088>.

They conclude that the hummingbirds occupy a larger portion of the colour space of birds than all other species combined (note that not all bird species were measured, the estimate of the total gamut of colours in birds was based on a sample of 111 species). The high colour diversity of hummingbirds is probably due to the fact that one of their main mechanisms of colour production is barbule structural colours, which can in theory produce nearly any hue and the fact that sexual selection is strong in this family (all or nearly all species have female-only parental care with polygynous social mating system).

This is a very interesting manuscript and while I think that the main conclusion is going to be supported over time, I have a few concerns that need to be addressed as detailed below:

Main concerns

1) Authors should be a bit more circumspect when calling hummingbirds the most colour-diverse family. The fact that sampling a fraction of hummingbirds species expands the colour gamut of all birds highlights that the latter estimate was not an accurate representation of the extent of colour diversity of birds. This means that further sampling may reveal other highly colour-diverse families. While I feel that it is likely to be true that hummingbirds shall be the most colour-diverse family, similar in-depth comparisons with other intensively sampled families where not attempted (eg sunbirds, parrots, tanagers). Hence, I suggest to qualify these statements throughout and potentially outline which other families could be contenders that should be assessed in a similar manner. This might prompt further exploration by other researchers (some of whom may already have the relevant data).

We have gone through the article and qualified these statements throughout. Additionally, the conclusion now includes a list of some of the other colorful avian families that would warrant exploration as well (Lines 322-326).

2) The use of convex hull volume to assess colour gamut has been criticised and alternative methods have been suggested (e.g. Delhey 2015 Sci Rep Gruson 2020 Methods Ecol Evol). It would be good to make use of these approaches (in particular the one outlined in Gruson 2020)

as alternative ways to quantify total colour space occupied. The main problem of convex hull volume is that it mostly contains empty space and is highly sensitive to outliers.

We are already aware of these issues because, as Gruson (2020) notes, Stoddard and Prum (2008) commented on them. We think that the results gained by using this new method will not contribute to our paper.

Gruson's method loses track of the fact that our goal is a quantification of achieved signal diversity in a sensory space. As quantifications of sensory experiences, the convex hull is *not* an exaggeration of the sensory diversity that a set of color points encompasses. All the edges of a convex hull are potentially observable color contrasts. Thus, from a sensory perspective, the convex hull is a precisely appropriate measure of the 3D span of a set of color points, regardless if some intervening points in the volume are unoccupied.

The reason to use a sensory space to quantify organismal coloration— rather than a PC-analysis of reflectance spectra, or a classification of pigment and nanostructural diversity, etc.— is that the sensory perception of the organisms themselves and the capacity for contrast among perceptions are central to the intellectual goal of quantification itself. In this important sense, using α -shapes would underrepresent the diversity of the perceivable color contrasts achieved by a set of color points.

Furthermore, it is not clear how to apply Gruson's method to a comparative question. Gruson's 'optimal' value of α is calculated based on user criteria and the dataset itself. Gruson (2020:959) writes, "There is no 'magic' value for α that will work for all datasets and this value can only be defined in the context of a given set of points... Users should feel free to adopt their own conditions and α values..." (Gruson 2020:959). Gruson makes no statement about what the appropriate α value should be when comparing *two* different datasets, which are very unlikely to have the same α optimal value. Should we compare α volumes based on different optimal α values? Do we combine all the data and calculate a single optimal α value for both datasets? Larger values of α will always produce larger α volumes and are prone to bias according to sample size (with larger datasets having larger optimal α values), so comparing two color point distributions based on different α values could be an unfair comparison. We think it is beyond the bounds of this manuscript to expand the development of Gruson's method to compare α volumes from different datasets.

Additionally, the concern that outliers can lead to overestimation of color gamut size is unwarranted (see below).

3) Similarly, it would be good to carry out resampling procedures to assess how much specific species contribute to the values reported. This may allow to compute confidence intervals around the values reported.

We chose not to carry out any resampling procedures because estimating the sensory gamut of a diverse group of species is not like estimating a statistic from a random sample drawn from a population. Except for measurement error, all current color points in the gamut are real

evolutionary achievements of the clade. These color points are not samples from a stochastic distribution or values distributed around some mean (at least not a mean we are interested in). Rather they are the product of exceptionally diverse, contingent history of plumage color evolution.

Our data acquisition method was specifically designed to document the maximum amount of color diversity with the least amount of sampling effort. (Still, we did sample approximately a third of all hummingbird species!) This method can be expected to produce the maximum number of "outliers" because we were specifically targeting species with interesting and distinct coloration. Beyond measurement error (not the source of this question), the only source of uncertainty is whether the sample is large enough to capture all the diversity of the clade.

Of course, it would be meaningful to investigate the color disparity of specific genera and subclades, and the contribution of different species. However, that would require complete species and plumage patch sampling, a well resolved phylogeny, and lineage-focused methods of analysis (rather than phylogenetic regression, etc.).

4) As far as I understand the methods used to collect reflectance spectra differ between this manuscript and Stoddard & Prum 2011. I think that it would be good to add a bit of discussion on the potential for this to affect conclusions.

Yes, the methods do differ slightly. We mention these differences in the results section "Species Color Gamuts," but now have included it in the discussion as well (Lines 209-216).

*Specific minor comments (ln = line number)
Ln 39 complement*

Corrected.

Ln 105 how were melanin patches defined?

We defined melanin patches as "phaeomelanin or eumelanin patches with absence of structural coloration." This definition has been added to that section (Lines 108-109).

Ln 131 as currently known

Corrected.

Ln 161 why 'ultra-red'? do you mean ultraviolet-red?

Yes. We have clarified this (Line 166).

Ln 174 true purple?

True purple is defined as colors created by spectra with peaks in the blue and red wavelengths. We have clarified this in the article (Line 129)

Ln 188 densely, how did you define this?

Density refers to the proximity and overlap among color points over regions of color space, as shown in Figure 3d.

Ln 193 should a statement like this not be accompanied by some estimate of evolutionary rates? A specific clade may be able to evolve a large colour gamut simply because it had a long time to do so.

We have added this observation. Based on McGuire et al (2014), the 336 species of hummingbirds have radiated over the past 22 million years (Lines 219-222).

Ln 194-196 I think that some further elaboration as to what mechanism determines the difficulty of evolving/producing such colours would be good here.

We have added the remark that, although the parameters are understood, that little is known about the evolutionary mechanisms of barbules structural colors (Lines 198-200).

Ln 197-200 the argument being that uv+green colours are not very conspicuous (because they match the green of leaves) nor very cryptic (because they have the UV component that is not so marked in leaves)? Because otherwise you could assume that green colours could evolve because they are cryptic in a vegetated environment. Many green colours that are meant to be cryptic do have a UV component (e.g. parrot green). It might be interesting to compare how well/bad these green colours match the spectrum of green leaves.

We cut this speculative comment.

Ln 205-206 +time available.

We have added this observation (see above; Lines 219-222).

Ln 254 revise writing

We cut this section and incorporated it into the paragraph above. (Lines 252-272)

Ln 267-271 do you think that this would be an example of matching between sensory system and signal as predicted by the sensory drive theory? Perhaps some mention would be required, plus some further ideas on how to test this. Previous studies in birds are a bit equivocal.

Not quite. Sensory drive proposes that natural selection on sensory systems (usually due to habitat and habitat structure) produce coevolutionary changes in social signals. Ours is really an aesthetic hypothesis— hummingbird plumages have evolved to produce the most saturated colors

possible given their specific visual system. This is an aesthetic (sexual) selective drive given their visual system.

Ln 288-290 some further discussion of the ideas outlined in Wilkins & Osorio 2019 J exp Biol would be good here.

We chose not to include a discussion of this paper because we thought it was not relevant given that it did not use of a tetrahedral color space model. Also, we note that Wilkins and Osorio did not cite any tetrahedral color space literature in their paper. If there were notable disadvantages of this common method, it seems likely that they would have raised them.

Ln 311 as assessed by human observers.

We added this to make it clearer (Line 335).

Reviewer #2 (Remarks to the Author):

The purpose of this research manuscript was to explore the plumage color gamut of the hummingbird family (Trochilidae) based on 114 species of hummingbirds. This research project builds on the work by Stoddard and Prum (2011) that had previously established the avian color gamut for 111 species of birds representing the known diversity of avian color production mechanisms. Stoddard and Prum (2011) observed structural colors made the largest contribution to the avian color gamut and predicted that greater sampling of structural colors would further expand the avian color gamut. This manuscript specifically tests this prediction with hummingbirds, which are known for their diverse and complex barbule structural color. In addition to this, the manuscript also further expands on the work of Stoddard and Prum by examining what constraints avian coloration mechanisms have on the size of the avian color gamut.

The authors found that the structural colors of hummingbirds occupy 34.2% of the color space perceived by birds and demonstrate that the hummingbird structural color gamut expands on the previously established space avian color gamut determined by Stoddard and Prum (2011). The increases to the avian color gamut are all due to contributions of hummingbird structural colors that were not found in the taxa used in the original study. These results provide further support for the hypothesis that barbule structures impose the fewest constraints on the diversification of plumage coloration. The authors also determined that the size of the gamut volumes associated with specific plumage patches were related to differences in their life history roles.

This study is an important contribution to expanding our knowledge of the relationship between the avian visual system and the coloration expressed by species of birds. The original study by Stoddard and Prum (2011) provided a preliminary overview of the avian color gamut using a relatively small sample of species (compared to the total number of known avian taxa) representing a diverse range of taxa from several different orders. The current study is the next step in getting a closer look at the color gamut within a family of closely related species that exhibit diverse coloration now that the method and tools have been established. Of all of the

avian species, hummingbirds would be the family to further test the contributions of structural colors to the avian color gamut. I believe the sample size and the choice of taxa used was appropriate for the hypotheses tested in this study and the data collection was thorough, with repeated measures of each individual for each of the six standardized patches. The methods outlining the spectra measures are detailed and well supported in the literature. The authors do an excellent job explaining the color gamut of the hummingbird structural colors in terms of their contributions to the total avian color gamut and also look closely at the color gamuts of specific species that expanded the volume of the avian color gamut. The figures are informative and easy to understand. I also believe the discussion is well-written. The explanations for the potential physiological and structural mechanisms behind why hummingbirds are so colorful are detailed and support the results reported in the study.

We appreciate these comments.

My only issue is that I feel that the discussion ends abruptly without any discussion of next steps, and particularly whether there is a need to investigate the plumage coloration and mechanisms of color in other families of birds with these fine-scale analyses to investigate the avian plumage color gamut further.

We added a new paragraph about other colorful families that could rival the gamut of hummingbirds (Lines 322-326).

Below, I list specific comments I have for the authors:

Lines 87-88: You say in the methods that 114 species from 68 genera represent 33% of extant hummingbird species, but I think this needs to be included in the main text here along with whether or not you have at least one representative of each of the current known genera. I think this is important because one of the main goals of the project is to expand on the sampling done for structural colors in the original Stoddard & Prum (2011) data set.

Stoddard and Prum (2011) examined three species of hummingbirds from three genera. We have made this clearer (Line 75), and also referred to the percent of extant hummingbird genera in our sample (~ 60%; Line 90).

Line 109: It would be useful to know how many species of hummingbirds were included in these previous analyses of avian color diversity so that it is clearer how this study has improved upon the sampling in these previous studies.

See above.

Line 131 & Line 146: I think this should be reworded to say so that it states that hummingbirds are the most diversely colored family of birds in the world based on the current data set. One aspect that is not clear in the paper is if this type of analysis has been done for any other bird family other than hummingbirds. I realize that to the human eye hummingbird coloration may look the most diverse, but this claim in line 131 is hard to justify without far more data on other

families of birds to match the amount of data that we now have on hummingbirds. In line 147 you do state that hummingbirds contain nine of the most diversely colored species of known birds, but this needs to be clarified in terms of describing families.

We have qualified our claims and outlined what future families should be studied in our concluding paragraph (Lines 322-326).

Line 190: You need a reference here demonstrating that throat and crown plumage patches play important roles in hummingbird communication.

Done (Line 195).

Line 257: I don't understand this heading what do you mean by larger vs. hummingbird color gamut size? Based on the text in the section below it seems as though you are comparing the results in different visual systems and the subtitle should reflect that.

We meant to write "Larger VS-sensitive Cone Type Gamut" and have updated the heading (Line 274).

Lines 292-303: The last paragraph here ends the paper abruptly. You imply that you believe having an ancestor that was nocturnal may contribute to the diversity of structural coloration observed in hummingbirds, but I'm not clear why you predict this would be the case. I would like to see the end of this paper in a few sentences focus on next steps; particularly whether or not there is a need to investigate the plumage coloration and mechanisms of color in other families of birds with these fine-scale analyses to investigate the avian plumage color gamut further. This is an impressive amount of work, but the question is does it need to be done in other families or is this a good stopping point for our understanding of the avian color gamut?

We have added references documenting that an evolutionary history of nocturnality often affects the visual system of organisms in leading to the loss or degradation of certain cone cell types (Lines 314 - 321).

We have also added a new paragraph to suggest similar research on other families that may rival hummingbirds in color diversity (Lines 322-326).

Line 312: Is clade the same as your 68 genera included in line 88? Does this mean you have a representative species from each currently known genera of hummingbirds? I think this needs to be clarified.

By clade, we are referring to the nine major hummingbird clades as defined by McGuire (Line 336). This included approximately 60% of extant hummingbird genera.

References:

Stoddard, M. C. & Prum, R. O. How colorful are birds? Evolution of the avian plumage color gamut. Behavioral Ecology 22, 1042-1052 (2011).

Reviewer #3 (Remarks to the Author):

I have serious concerns about this work, both conceptual and methodological. First, from the title it may be inferred that the color gamuts of all species of birds, or a significant proportion of species, have been investigated and compared with those of hummingbirds. However, the only reference study is Stoddard and Prum (2011), who measured the amplitude of the color gamut of 111 species of birds. Far from being a significant representation of all extant birds.

We never implied that it was based on a large sample of avian species, but this study was based on a complete sample of all known avian plumage color mechanisms, and thus represents a significant estimate of the gamut achieved by all birds.

The conclusion of the present study that hummingbirds have the greatest plumage color diversity of all birds and that this represents a considerable expansion of the avian color gamut as previously estimated, thus, is not fully justified. It may be an expansion of the color gamut provided by the previous study of the team (Stoddard and Prum), but not of the known avian color gamut as stated.

We have modified those claims throughout the manuscript.

My second concern refers to how color is considered and measured here. Color is not only a neural process output in the receivers. Color is, firstly, a physical property of the objects. It is thus an intrinsic property of the objects, not only something that occurs in the brain of the animals that perceive it. I recommend the authors to read the works by Francisco J. Varela on this topic. As a consequence, color should not be measured by tetrahedral or any color space models. To be objective and obtain results that are comparable among taxa (how could we compare the results of this article with the color gamut of mammals, for example?), color diversity should be quantified by studying variability in the physical phenomenon, i.e. the electronic spectra.

The point of investigating color diversity in taxon-specific color spaces is that color perception is a subjective experience. This paper is about the evolution of color diversity as perceived by the birds themselves– i.e. the potential influence of color perception on color evolution. Developing an objective universal method for the comparison of subjective perceptual spaces is outside of the bounds of this study.

It is also surprising that the measurement of color in hummingbirds was done in an standardized way considering the same plumage patches in all species, but then, if there were other patches with a color that was distinctive to the eyes of the authors, these were also measured. This is equivalent to measure the colors that the authors were perceiving on birds. The amplitude of the color gamut is therefore constrained by the human perception of distinct colors, as the colors that were measured depended on what the authors perceived.

It is true that we don't have any measures of plumage patches that we didn't measure. But the point of our sampling was to explore the gamut as much as we could. It would not have been practical to keep measuring all 3mm² spots on the plumage of all specimens in search of colors that are different but do not appear differently to human eyes.

Lastly, the discussion is exceedingly long and speculative. For example, there are many mentions to variability in morphological features of barbules causing differences in color, but no morphological measures are provided in this study.

We note that two other reviewers requested additions to the discussion. This is obviously not a morphological study. However, many papers on the morphology of hummingbird papers were cited.

Reviewers' comments:

Reviewer #1 (Remarks to the Author):

The other reviewers and I made a series of constructive suggestions on the manuscript by Venables et al. that attempts to quantify the extent of colour diversity within hummingbirds which, according to our visual perception may be one of the most colourful bird families.

Unfortunately, the authors chose not to really engage with several of my suggestions. Below I briefly elaborate.

1) I suggested to add alternative ways to estimate the occupied colour volume. The authors dismissed the suggestion claiming to be aware of the limitations of the methods but not really explaining them in the manuscript. I would have expected at least to have seen some thought devoted to these issues in the manuscript. The authors do use a few other metrics such as colour span or hue disparity, but they only report results for colour volumes. It is unclear (and not explained) why these other metrics were computed at all. I note that colour span is less sensitive to outlying points and not necessarily correlated with sample size (as convex hull volume is).

2) I suggested to use a resampling procedure to try to estimate the uncertainty around these estimates. The authors respond that this is not necessary, because they are interested in the colour range. However, they do find that the previous estimate of colour gamut (which includes a much more taxonomically diverse sample of species encompassing all known mechanisms of colour production) is clearly an underestimate. This indicates that the previous estimate is inadequate and incomplete. Something similar may be happening to their current assessment of hummingbird colour diversity. Convex hull volumes always increase with more species in a sample, so the fact that they have more measurements for hummingbirds than for the other species already biases the sample towards the key focus group. Providing error estimates around these numbers, by, for example, sampling a similar number of species or data points and comparing both volumes may have been a simple alternative. Contrary to what the authors state, measuring reflectance spectra has an inherent level of error depending on how the probes are held, how strongly they are pressed against the plumage, the state of the specimen being measured, etc. This may be more of an issue for the Stoddard & Prum dataset, which takes me to the next comment.

3) I indicated the necessity of more fully addressing the issue that differences in measuring protocol may also bias conclusions. Unlike the Stoddard and Prum dataset here the authors took extra care to find the most saturated reflectance spectra (as assessed by human observers). This difference was only cursorily acknowledged and the authors state that this difference should not concern the entire gamut. I do not quite follow why this would be the case. They mention their large sample size, but this is only true (to some extent) for the hummingbird dataset, again biasing the outcomes.

4) Lines 323-326 I would suggest that you cite here previous studies that have measured many species belonging to these (and other colourful) families, notably:
Price-Waldman, Rosalyn M., Allison J. Shultz, and Kevin J. Burns. "Speciation rates are correlated with changes in plumage color complexity in the largest family of songbirds." *Evolution* 74.6 (2020): 1155-1169.
Shawkey, M.D., Igic, B., Rogalla, S., and Goldenberg, J. (2017). Beyond colour: consistent variation in near infrared and solar reflectivity in sunbirds (Nectariniidae). *Sci. Nat.* 104, 78.
Cooney, Christopher R., et al. "Sexual selection predicts the rate and direction of colour divergence in a large avian radiation." *Nature communications* 10.1 (2019): 1-9.
Doucet, Stéphanie M., Daniel J. Mennill, and Geoffrey E. Hill. "The evolution of signal design in manakin plumage ornaments." *The American Naturalist* 169.S1 (2007): S62-S80.
Merwin, Jon T., Glenn F. Seeholzer, and Brian Tilston Smith. "Macroevolutionary bursts and constraints generate a rainbow in a clade of tropical birds." *BMC evolutionary biology* 20.1 (2020): 1-19.
Ligon, Russell A., et al. "Evolution of correlated complexity in the radically different courtship signals of birds-of-paradise." *PLoS biology* 16.11 (2018): e2006962.

Reviewer #2 (Remarks to the Author):

After reading the revised manuscript, I believe the authors have thoroughly addressed my questions and concerns from the original draft of the manuscript. As I said in my first review, I think paper is an important contribution to expanding our knowledge of the relationship between the avian visual system and the coloration expressed by species of birds. I also believe it provides novel insight into the evolution of structural coloration in birds. I would recommend the revision be accepted for publication.

Reply to Reviewers:

Reviewers' comments:

Reviewer #1 (Remarks to the Author):

The other reviewers and I made a series of constructive suggestions on the manuscript by Venables et al. that attempts to quantify the extent of colour diversity within hummingbirds which, according to our visual perception may be one of the most colourful bird families.

Unfortunately, the authors chose not to really engage with several of my suggestions. Below I briefly elaborate.

In our previous response to this reviewer, we detailed why we chose not to incorporate all of Reviewer 1's suggestions, many of which this reviewer has repeated in this current review without addressing our comments specifically. However, at the suggestion of this reviewer, we did conduct the resampling analysis (see below), which supports the points we made in our previous response.

1) I suggested to add alternative ways to estimate the occupied colour volume. The authors dismissed the suggestion claiming to be aware of the limitations of the methods but not really explaining them in the manuscript. I would have expected at least to have seen some thought devoted to these issues in the manuscript. The authors do use a few other metrics such as colour span or hue disparity, but they only report results for colour volumes. It is unclear (and not explained) why these other metrics were computed at all. I note that colour span is less sensitive to outlying points and not necessarily correlated with sample size (as convex hull volume is).

In our previous response to this reviewer, we detailed why we chose not to use the other methods they suggested. To repeat in brief, the reviewer's concerns for the influence of "outliers" is non-appropriate in the description of the "achieved gamut of a clade"— a macroevolutionary concept which should not be estimated like the mean of a random sample of a population.

As we stated previously, we do not adopt either Gruson's alpha-shapes or Delhey's color space grids as alternative estimates of color space occupancy because we think that the original concept of the color gamut, as proposed by Stoddard and Prum, is a perceptual concept for which the convex hull is actually the appropriate measure. It appropriately includes all the "edges" between the distribution of color points which are observable color contrasts. The fact of unevenness in distribution of color points in color space may be an interesting question for other research programs, but it is not an issue for the estimation of a color gamut. In response to this reviewer, however, we have now added a paragraph at the end of our Methods section (lines 421-433) that is similar to our previous response and that above.

The other metrics (color span etc.) were included because they were used previously by Stoddard and Prum (2011) and this facilitates comparison of our results to theirs. In this revision, we further clarified this in lines 412-413.

2) I suggested to use a resampling procedure to try to estimate the uncertainty around these estimates. The authors respond that this is not necessary, because they are interested in the colour range. However, they do find that the previous estimate of colour gamut (which includes a much more taxonomically diverse sample of species encompassing all known mechanisms of colour production) is clearly an underestimate. This indicates that the previous estimate is inadequate and incomplete. Something similar may be happening to their current assessment of hummingbird colour diversity. Convex hull volumes always increase with more species in a sample, so the fact that they have more measurements for hummingbirds than for the other species already biases the sample towards the key focus group. Providing error estimates around these numbers, by, for example, sampling a similar number of species or data points and comparing both volumes may have been a simple alternative. Contrary to what the authors state, measuring reflectance spectra has an inherent level of error depending on how the probes are held, how strongly they are pressed against the plumage, the state of the specimen being measured, etc. This may be more of an issue for the Stoddard & Prum dataset, which takes me to the next comment.

In this revision, we conducted a resampling analysis of our data as the reviewer requested using a rarefaction scheme. Our results confirm our previous statements. We resampled datasets of 60, 70, 80, 90, 100, and 110 species without replacement 10 times each, and calculated the average and standard deviation in plumage color gamut volume. Gamut volume exhibited minimal increases in volume with increasing number of species in datasets of ≥ 90 species (Supplementary Figure 1). Additionally, the standard deviation of total hummingbird gamut size became increasingly negligible as sample size approached that of our study, i.e. 114 species (coefficient of variation = 0.0455 at 110 species; Supplementary Table 1 and Supplementary Figure 1; copied below). Notably, our resampling showed that the VS hummingbird color gamut on average exceeded that of the previous estimate of all birds even when only including 60 species of hummingbird. Given the results of our resampling, we believe that our sample of species is neither inadequate nor incomplete.

Thus, contrary to this reviewer's statements above, the convex hull *does not* always increase as the number of samples gets larger. (Our estimates of the hummingbird color gamut converged on our final value for sample above 90 species.) Color volumes increase if birds with more distinct plumage colors are included, as our resampling analysis shows. Further, Stoddard and Prum's previous estimate was not an underestimate simply because they did not include a large enough sample size. Their sample also failed to include enough biodiversity- taxa using different plumage color mechanisms.

Stoddard and Prum measured spectra from 111 species of birds, which is almost the same number of species as we measured (114) and yet the VS hummingbird gamut is almost one third bigger than that found by Stoddard and Prum ($7.41E-2$ vs $5.62E-2$). A sample with more species and/or more measurements would be more likely to be larger if samples are chosen randomly from a normal distribution. But biodiversity is not distributed like this, and it is inappropriate to investigate macroevolutionary concepts- like a color gamut- as if it were. Both we and Stoddard

and Prum chose species by *looking for* outliers, which is the appropriate way to estimate the diversity achieved by a clade.

We address the reviewer’s comment about differences in measuring protocol in the response to the comment below.

Supplementary Table 1: Means, standard deviations (St.dev), and coefficients of variation (CV) in hummingbird plumage color gamut size for the resampled datasets according to number of species (sps) resampled.

	60 sps	70 sps	80 sps	90 sps	100 sps	110 sps
Mean	0.0627	0.0640	0.0658	0.0711	0.0718	0.0723
St.dev	0.0057	0.0067	0.0061	0.0034	0.0032	0.0033
CV	0.0908	0.1052	0.0934	0.0477	0.0450	0.0455

Supplementary Fig. 1: The relationship between the number of species included in a dataset and hummingbird plumage color gamut size. The red triangles are the means of the ten resampled datasets corresponding to that number of species, which are the black points. The blue line shows the volume we calculated with our total hummingbird data. Values for the means, standard deviations, and coefficients of variation are located in Supplementary Table 1.

3) I indicated the necessity of more fully addressing the issue that differences in measuring protocol may also bias conclusions. Unlike the Stoddard and Prum dataset here the authors took extra care to find the most saturated reflectance spectra (as assessed by human observers). This difference was only cursorily acknowledged and the authors state that this difference should not concern the entire gamut. I do not quite follow why this would be the case. They mention their large sample size, but this is only true (to some extent) for the hummingbird dataset, again biasing the outcomes.

In our revision, we have made it clear how our sampling method differs from Stoddard and Prum and why. Given the optical properties of iridescent colors, we used methods that will produce a more accurate measurement of the diversity of observable colors produced by these plumages. This was not the primary goal of Stoddard and Prum's design. Although our methods are "biased" toward greater variation, they are necessary for an accurate estimate of the diversity of plumage colors within the hummingbirds, and therefore birds as a whole.

We have included these statements specifically in this manuscript (lines 220-231).

4) Lines 323-326 I would suggest that you cite here previous studies that have measured many species belonging to these (and other colourful) families, notably:

Price-Waldman, Rosalyn M., Allison J. Shultz, and Kevin J. Burns. "Speciation rates are correlated with changes in plumage color complexity in the largest family of songbirds." *Evolution* 74.6 (2020): 1155-1169.

Shawkey, M.D., Igc, B., Rogalla, S., and Goldenberg, J. (2017). *Beyond colour: consistent variation in near infrared and solar reflectivity in sunbirds (Nectariniidae)*. *Sci. Nat.* 104, 78.

Cooney, Christopher R., et al. "Sexual selection predicts the rate and direction of colour divergence in a large avian radiation." *Nature communications* 10.1 (2019): 1-9.

Doucet, Stéphanie M., Daniel J. Mennill, and Geoffrey E. Hill. "The evolution of signal design in manakin plumage ornaments." *The American Naturalist* 169.S1 (2007): S62-S80.

Merwin, Jon T., Glenn F. Seeholzer, and Brian Tilston Smith. "Macroevolutionary bursts and constraints generate a rainbow in a clade of tropical birds." *BMC evolutionary biology* 20.1 (2020): 1-19.

Ligon, Russell A., et al. "Evolution of correlated complexity in the radically different courtship signals of birds-of-paradise." *PLoS biology* 16.11 (2018): e2006962.

We have added these references to those lines (now lines 338-341).

Reviewer #2 (Remarks to the Author):

After reading the revised manuscript, I believe the authors have thoroughly addressed my questions and concerns from the original draft of the manuscript. As I said in my first review, I think paper is an important contribution to expanding our knowledge of the relationship between the avian visual system and the coloration expressed by species of birds. I also believe it

provides novel insight into the evolution of structural coloration in birds. I would recommend the revision be accepted for publication.

We were happy to see that Reviewer 2 thought this was an important paper with novel insights.

REVIEWERS' COMMENTS:

Reviewer #1 (Remarks to the Author):

The authors have addressed my main concerns. Two minor points that the editor may or may not want to enforce:

Ln 48 not 'all' but a sample.

Ln 341 the authors did not refer to data available for the family Pipridae (Doucet, Stéphanie M., Daniel J. Mennill, and Geoffrey E. Hill. "The evolution of signal design in manakin plumage ornaments." *The American Naturalist* 169.S1 (2007): S62-S80.) even though they say they do so in the response letter.

Reply to Reviewers:

We thank the reviewers for all their feedback. We have incorporated all the minor edits that they requested.

REVIEWERS' COMMENTS:

Reviewer #1 (Remarks to the Author):

The authors have addressed my main concerns. Two minor points that the editor may or may not want to enforce:

Ln 48 not 'all' but a sample.

We addressed this line to say "a comprehensive sample of all" (Line 58).

*Ln 341 the authors did not refer to data available for the family Pipridae (Doucet, Stéphanie M., Daniel J. Mennill, and Geoffrey E. Hill. "The evolution of signal design in manakin plumage ornaments." *The American Naturalist* 169.S1 (2007): S62-S80.) even though they say they do so in the response letter.*

We have added this reference (Line 423).